# Heterogenic Origin of Micro RNAs in Atlantic Salmon (*Salmo salar*) Seminal Plasma

**DOI:** 10.3390/ijms21082723

**Published:** 2020-04-15

**Authors:** Teshome Tilahun Bizuayehu, Igor Babiak

**Affiliations:** Genomics Group, Faculty of Bioscience and Aquaculture, Nord University, 8049 Bodø, Norway; teshome.bizuayehu@uib.no

**Keywords:** Atlantic salmon, gene expression, microRNA, ovary, seminal plasma, spermatogenesis, testis

## Abstract

The origin and contribution of seminal plasma RNAs into the whole semen RNA repertoire are poorly known, frequently being overlooked or neglected. In this study, we used high-throughput sequencing and RT-qPCR to profile microRNA (miRNA) constituents in the whole semen, as well as in fractionated spermatozoa and seminal plasma of Atlantic salmon (*Salmo salar*). We found 85 differentially accumulated miRNAs between spermatozoa and the seminal plasma. We identified a number of seminal plasma-enriched and spermatozoa-enriched miRNAs. We localized the expression of some miRNAs in juvenile and mature testes. Two abundant miRNAs, miR-92a-3p and miR-202-5p, localized to both spermatogonia and somatic supporting cells in immature testis, and they were also highly abundant in somatic cells in mature testis. miR-15c-5p, miR-30d-5p, miR-93a-5p, and miR-730-5p were detected only in mature testis. miRs 92a-3p, 202-5p, 15c-5p, and 30d-5p were also detected in a juvenile ovary. The RT-qPCR experiment demonstrated lack of correlation in miRNA transcript levels in seminal plasma versus blood plasma. Our results indicate that salmon semen is rich in miRNAs, which are present in both spermatozoa and seminal plasma. Testicular-supporting somatic cells are likely the source of seminal plasma enrichment, whereas blood plasma is unlikely to contribute to the seminal plasma miRNA repertoire.

## 1. Introduction

Small RNAs are involved in the regulation of various genetic elements important for differentiation, proliferation, and apoptosis. Mature gametes carry the parental information, which regulates early embryonic development [1]. This parental information includes paternal epigenetic patterns and elements [2,3,4], delivered through spermatozoa, which regulate embryonic transcription [5,6], and thereby imprint the fate of an embryo. The sperm contains different classes of RNAs, but very little is known about their functions. Sperm RNAs can enter the oocyte during fertilization [7], and studies on *Xenopus* sp. and mammals suggest their functional relevance [3,6]. Several hypotheses on the role of sperm RNAs in embryonic development have been formed [8,9], yet these conjectures need further experimental validation.

Teleost fishes are the largest group of vertebrates, and their predominant reproductive features, such as a lack of accessory glands in semen production, external fertilization, or spermatozoa activation upon contact with water [10], make them distinct from tetrapods. Virtually, nothing is known about the potential transfer and possible role of paternal RNAs in embryo formation in fish.

The whole semen RNAs are found not only in germ cells (spermatozoa) but also in the seminal plasma, where they are bound in protein complexes and encapsulated in microvesicles [11,12]. Seminal plasma is an important constituent of semen and has a vital role in spermatozoa metabolism, survival, and motility. It contains inorganic and organic compounds, proteins, and RNAs [13]. In fish, seminal plasma has been implicated in osmotic balance, proteolytic activities, and fertilization success [14,15,16]. However, the repertoire, abundance, origin, and functions of seminal plasma RNAs remain unknown in fish.

Micro RNAs (miRNAs) are among the RNAs that are present in the sperm of vertebrates. These non-coding RNAs, approximately 22 nucleotides long, function mainly in translational suppression by binding at 3′ UTR of mRNA to regulate various biological processes, including spermatogenesis [17]. In mouse, ingression of sperm miRNA to oocytes causes heritable epigenetic change in gene expression [18]. In addition, a sperm-borne miRNA miR-34c is essential for the first cell division [2]. miRNA expression has been characterized in the testes of some teleost fish [19], such as Atlantic halibut *Hippoglossus hippoglossus* [20], Atlantic cod *Gadus morhua* [21], yellow catfish *Pelteobagrus fulvidraco* [22], or Nile tilapia *Oreochromis niloticus* [23]; however, the testis is a heterogenic organ composed of germline and somatic cell lineages, and the cellular origin of miRNAs has not been determined. Whole semen [24] and spermatozoal [25] miRNA profiling has been performed in zebrafish (*Danio rerio*), confirming the abundance of miRNAs in fish sperm.

Quite often, the reported studies on sperm RNA refer to germ cell/spermatozoa RNA in reasoning and conclusions, while they have actually been performed on the whole semen. This is confusing, because no account is taken on possible non-spermatozoal RNA’s contribution in the whole semen. The objective of the present study was to profile miRNA transcriptomes in fish semen, fractionated to spermatozoa and seminal plasma, in order to determine the contribution and content of non-spermatozal miRNAs in the semen. We chose Atlantic salmon (*Salmo salar*) as a model because of its relatively large body size and resulting collectible volumes of semen and blood plasma. To elucidate the possible origin of seminal plasma miRNAs, we performed in situ hybridization of chosen differentially accumulated miRNAs in the testis, and another experiment comparing the chosen miRNAs in semen fractions with blood plasma miRNAs.

## 2. Results

### 2.1. Atlantic Salmon Sperm Small RNA Sequencing and Annotation

On average, we obtained 31.6 million sequences per library and the total number of sequences exceeded 284 million. Of these, 53.8% mapped to the Atlantic salmon genome without a mismatch. The length distribution showed that over 44% of the sequences were in the size range of 42–43 nucleotides (nts) (Figure 1A). In seminal plasma, some enrichment was found for the size range 27–35 nts compared to spermatozoa (34.6% versus 17.9%), whereas the fraction of the size range 15–26 nts constituted 12.1% and 25.2%, in seminal plasma and spermatozoa, respectively (Table 1 and Figure 1A). In total, we found 30,205,087 unique sequences comprising 2.9% miRNAs, 9% piRNAs, 25.8% tRNA fragments (tRFs), and 17.4% mRNA fragments (Figure 1B).

### 2.2. miRNA Profiling in the Whole Sperm and Its Fractions

In total, we identified 306 mature miRNAs representing 97 families (Appendix A). The highest average number of reads and miRNA diversity was found in the seminal plasma (Figure 1C, Appendix A, and Appendix A). There was a high degree of similarity in miRNA types and their abundances between the whole semen and spermatozoa (*r* = 0.97), whereas the correlation was considerably lower (*r* = 0.61) between the seminal plasma and spermatozoa, as well as seminal plasma and the whole sperm (Appendix A). The cluster analysis confirmed the separate clustering of seminal plasma samples versus the whole semen and spermatozoa samples (Figure 2).

Sixteen miRNA families contributed to over 94% of all miRNA reads; out of them, four families (miR-15, miR-17, miR-25, and miR-30) constituted approximately 68%, 50%, and 65% of the total read counts for spermatozoa, seminal plasma, and the whole sperm, respectively (Figure 3). Notably, miR-202 constituted approximately 16% of all miRNA reads in the seminal plasma, whereas it was showed negligible levels in the spermatozoa fraction.

We observed variation in the normalized read counts among individuals. Highly pronounced variability (> 50,000 reads-per-million, rpm) was found in miR-25-3p, miR-30b-5p, and miR-92a-3p in spermatozoa fraction. In the seminal plasma samples, miR-30b-5p, miR-92a-3p, and miR-202-5p accumulation was considerably variable among males (Appendix A).

The differential expression was profound between the seminal plasma and spermatozoa, with 85 differentially accumulated miRNAs (Figure 4). They represented 25.2% and 25.5% of the total miRNA read counts in the spermatozoa and seminal plasma, respectively. Among the top 10 dominant miRNAs, five miRNAs (miR-15c-5p, miR-16a-5p, miR-26a-5p, miR-30b-5p, and miR-92a-3p) did not show any significant differences between the semen fraction samples (Figure 4). Some miRNAs, such as miR-730-5p, showed relatively higher accumulation in spermatozoa compared to seminal plasma (Figure 4).

### 2.3. Spatial Expression of Selected miRNAs in Gonads

Localization of selected differentially accumulated miRNAs in the testis was performed in order to determine whether the differential abundance between the sperm fractions was associated with the cellular origin of miRNAs. The positive control, U6 snRNA (a housekeeping gene used as a positive control in miRNA in situ hybridization), showed signal in the nuclei of germ cells and somatic supporting cells in a juvenile testis (Appendix A), whereas in a mature testis, the signal was strong in somatic supporting cells but poorly distinguishable in spermatozoa due to their small size (Appendix A). No signal was detected in negative controls, where a scramble miRNA probe was used (Appendix A). In juvenile testis, the two dominant miRNAs, miR-92a-5p and miR-202-5p, were localized in spermatogonia and somatic supporting cells surrounding lobules (Figure 5A). In addition, miR-202-5p was found in primary spermatocytes, and its concentration in the space surrounding spermatogonia was prevalent (Figure 5A). No signal was detected for miR-15c-5p, miR-30d-5p, and miR-93a-5p (Figure 5A), whereas miR-730-5p produced a faint signal (Figure 5A). In mature testis, lobules were broken, and spermatozoa were released to lumens surrounded by somatic supporting cells forming ridge-like structures (Figure 5B). Given the fact that the U6 snRNA positive control signal was poorly distinguishable in spermatozoa (Appendix A), in situ hybridization (ISH) of sections of mature testes provided information primarily on the signal intensity in somatic supporting cells, which was strong for miR-30d-5p, miR-202-5p, and miR-92a-5p, rather weak for miR-15c-5p, and poorly detectable for miR-93a-5p and miR-730-5p (Figure 5A).

To investigate cross-sex conservation of the regulation of gonad development, we localized miRNAs in a juvenile ovary of Atlantic salmon. The U6 snRNA control gave clear signals in the nuclei of germ cells and somatic cells. Similar to the juvenile testis, the ISH showed expression of miR-92a-5p and miR-202-5p in both previtellogenic oocytes and somatic supporting cells; the latter one was massively abundant. Signals for miR-15c-5p and miR-30d-5p were clear in oocytes but poorly distinguishable in somatic supporting cells, while miR-93a-5p and miR-730-5p expression was not distinguishable (Appendix A).

### 2.4. Comparison of Sperm and Blood Plasma miRNAs

To check whether the seminal plasma-enriched miRNAs originated from blood, we performed an additional RT-qPCR experiment. We found, in accordance with RNA-seq data, clear enrichment of the tested miRNAs in seminal plasma compared to the whole semen or spermatozoa, with the exception of miR-192, which was present in spermatozoa but nearly absent in the seminal plasma (Figure 6). We tested the expression of miR-451a as blood-specific miRNA in this experimental context. Indeed, blood plasma had a high accumulation of miR-451a; although it was not detected in our RNA-seq experiment, miR-451a was also moderately present in the seminal plasma. Aside from miR-451a, miR-204 was relatively abundant in the blood plasma, while miR-145-3p, miR-148a-3p, and miR-7 were accumulated jn low levels. None of the tested seminal plasma-enriched miRNAs showed similar abundance levels in the blood plasma.

### 2.5. GO Terms Annotation of the Differentially Expressed miRNA Targets

To inspect the function of the differentially expressed miRNAs between spermatozoa and seminal plasma, we collected Atlantic salmon 3’UTR sequences from Genebank, which is not exhaustive since the genome annotation is incomplete. Out of 127 differentially expressed miRNAs, 101 miRNAs had 721 predicted target genes with 1573 assigned gene ontology (GO) term annotations (Appendix A). The highest numbers of GO term annotation were cellular integral component of membrane, GTP binding, ATP binding, metal ion binding, and regulation of transcription. This suggests that spermatogenesis is subject to regulation by the miRNAs and miRNA targets.

## 3. Discussion

In general, Atlantic salmon sperm contains diverse types of small RNAs, including miRNAs, tRNA fragments, and piRNAs. In mature sperm, miRNAs were less abundant than piRNAs and the highest accumulation was observed for tRFs (25% of the total reads). Enrichment of tRFs and reduction of piRNA have been reported during sperm maturation in mammals [26,27]. Similarly, piRNAs cover 9% of our small RNA sequencing. In teleost, piRNAs are abundant in the testis and ovary mainly to protect the germline from transposon propagation [28]. Given that germ cells carry the genetic information for future generations and rapid evolution of the piRNA pathway in teleost [29], the role of recent whole genome duplication in this pathway needs further investigation, for which Atlantic salmon can be a good model.

## 3.1. miRNA Heterogeneity in Sperm Fractions

We demonstrated for the first time that miRNAs in teleost semen have heterogenic cellular origin. miRNAs are present in body fluids through several possible export mechanisms, including microvesicles, apoptotic bodies, or high-density lipoproteins; they can also be vesicle free [30,31]. Differential accumulation of a substantial number of miRNAs in spermatozoa versus seminal plasma (Figure 4) suggests that aside from germ cells, a considerable portion of semen miRNAs originates from non-germ cells. The very high correlation of the miRNA profiles between the whole semen and spermatozoa contrasted with the considerably lower correlations between the seminal plasma and both the whole semen and spermatozoa (Figure 2). This indicates that miRNA heterogeneity in the seminal plasma (Figure 3), as well as its diversity (Appendix A) are mostly hidden when non-fractionated semen is analyzed. This effect can be due to the lower concentration of miRNAs in the seminal plasma as compared to that in spermatozoa.

Fish testis has germinal and interstitial compartments, which are composed of connective tissue and Leydig cells. The connective tissue contains fibroblasts, immunological cells, collagen fibers, myoid cells, blood vessels, and nerve fibers [32]. If seminal plasma miRNAs originated from the blood, they would likely reflect blood plasma miRNA content. However, our RT-qPCR experiment demonstrates a lack of concordance between miRNA abundance profiles in seminal versus blood plasmas, with only 2 out of 13 miRNAs showing a relative abundance in the blood plasma similar to that in the seminal plasma (Figure 6). It suggests that the blood plasma is not the major source of the enrichment in seminal plasma miRNAs, which is in line with the existence of a blood–testis barrier in fish [33]. Concordance of the ISH signal intensity in supporting somatic cells (Figure 5) with the abundance of seminal plasma miRNAs (Figure 4), along with intensive apoptosis in testicular cells associated with the final maturation of spermatozoa, suggests that somatic supporting cells, along with the germ cells, are a considerable source of seminal plasma miRNAs.

The diversity of miRNAs in the seminal plasma was higher than that in the whole sperm and spermatozoa (Appendix A). This demonstrates that a certain contingent of low-abundance miRNAs is not captured in the whole-semen samples subjected to a standard-depth sequencing. Aside from sperm fraction-specific expression, the captured diversity in the seminal plasma samples could result from the freeze-drying step in the procedure, which enhanced the efficacy of RNA extraction from the seminal plasma, where RNAs are natively highly diluted. 

miR-202-5p, a conserved gonad-dominant miRNA in teleosts [20,25,34], was the most prominent quantitative contribution of seminal plasma-enriched miRNAs to the general pool of seminal miRNAs (Figure 3). In mammals, miR-202-5p is a Sertoli cell-specific miRNA [35,36,37]. In contrast, it co-localizes with germ plasm components in zebrafish, suggesting its germline specificity [34]. However, in a recent study on zebrafish, miR-202-5p was highly abundant in mature gonads, both testes and ovaries, but not in the released mature spermatozoa or eggs [25]. In the present study, miR-202-5p localized to both germ and somatic lineages in juvenile testes and ovaries of Atlantic salmon and was highly abundant in somatic supporting cells in mature testes (Figure 5 and Appendix A). In medaka, miR-202-5p is abundant in unfertilized oocytes and in the follicular cells of the ovary, and is essential for the regulation of oogenesis [38]. Together, these results suggest a conserved role of miR-202-5p in reproductive processes in teleost gonad development during gamete maturation, and indicate supporting somatic cell origin of miR-202-5p in a mature gonad.

The heterogenic origin of salmon seminal plasma miRNAs is the most important information from this study. It means that the whole semen miRNA repertoire is not fully representative for the spermatozoa. If this feature is extended to other teleost species, it indicates a possible source of bias in all the studies where spermatozoa were meant to be analyzed, but the whole semen was analyzed instead. For example, in a study on zebrafish spermatozoa [24] where whole non-fractionated semen was used instead of purified spermatozoa, the authors reported very high levels of miR-202-5p in spermatozoa, whereas in the present study, it was demonstrated that the seminal plasma, but not spermatozoa, is the source of this miRNA in the semen (Figure 4 and Figure 6).

To investigate whether the extracellular seminal plasma miRNAs have regulatory functions, or alternatively, whether they are just leftovers of the past regulatory processes, we constructed a minigene. However, both sense and antisense RNAs were degraded within the first moments of the incubation by unknown factors, possibly nucleases.

## 3.2. miRNA Accumulation in Atlantic Salmon Sperm

Conserved and specific miRNAs were previously characterized in Atlantic salmon [39,40,41] but not in gonads or gametes. Some of the dominant miRNAs in Atlantic salmon semen, such as let-7a, miR-21a, miR-25, miR-26a, miR-128, and miR-202, were found to be abundant in zebrafish whole sperm analysis [24], and let-7a was among the most abundant miRNAs in human semen exosome samples [42], indicating some functional conservation among divergent species. Abundant miRNAs reported in mouse, including miR-15c-5p, miR-16a-5p, miR-20a-5p, miR-30b-5p, miR-92a-3p, and miR-93a-5p, were also found in Atlantic salmon sperm in the present study, indicating the enrichment of sperm with diverse types of miRNAs. Among these, miR-20a-5p and miR-93a-5p were expressed highly in mouse spermatozoa compared to spermatogonia with a concomitant decrease in their predicted targets, signal transducer and activator of transcription 3 (*Stat3*) and phosphatase and tensin homolog (*Pten*) [43]. miR-202-5p was highly enriched in seminal plasma compared to spermatozoa, which is in line with Sertoli cell accumulation of this miRNA in human [35].

In contrast to mammals, the miR-34 family was not found in Atlantic salmon sperm. Out of 35 miRNAs identified in human sperm [44], 23 miRNAs had homologs in our dataset, including the miR-19, let-7, and miR-30 families. Some of these miRNAs have been associated with male fertility [35,45,46]. Although all top 26 miRNAs identified in zebrafish whole sperm [24] were also found in our dataset, the level of accumulation was dissimilar for some of the miRNAs; for example, miR-22 and miR-122 were among the abundant miRNAs in zebrafish, while they were less abundant in salmon sperm. Thus, it is important to characterize and profile small RNA constituents of various teleost fish to obtain unifying and discriminatory features to understand the role of miRNAs in semen.

A previous study showed that primary miRNAs (pri-mir-1181, pri-miR-3648, pri-miR-3687, pri-mir-663, and pri-mir-181c) are found in human testis and spermatozoa [4]. This indicates the enrichment of sperm not only in mature miRNAs but also in the preceding forms, which may be transferred to the oocyte and further processed. The role of miRNAs beyond spermatogenesis has been illustrated by miR-34c, which contributes to the first cleavage by regulating B-cell leukemia/lymphoma 2 [2]. The miR-34 family has been implicated in the development of bovine gametes and embryos [47]. The role of sperm-borne miRNAs in aquatic species with an external fertilization strategy is not known.

Spermatogenic cells strictly depends on their interaction with the somatic elements of the testis, which requires the expression of many genes encoding proteins with a role in epithelial transport. Many of the predicted targets of differentially expressed miRNA include several cation, anion, and solute transport-related processes; for example, Solute carrier family 41 member 2, vesicle transport protein SFT2B, and potassium channel subfamily K member 6, among others. Among several kinds of activating signals, ions play a major role in salmonid sperm activation [48]. Proteolysis pathways genes were predicted targets of differentially expressed miRNAs. In salmonids, proteasomes modulate the activity of outer arm dynein to activate dynein-driven microtubule sliding for sperm motility [49]. Genes that regulate sperm motility, such as cAMP-responsive element-binding protein-like 2, and ATP binding proteins were among the predicted targets. Previously, cAMP-dependent phosphorylation of axonemal proteins has been reported to regulate the motility of sperm in salmonid fish [50]. In general, miRNAs regulate a number of genes that are involved in the regulation of transcription, membrane transport, and ATP synthesis pathways, indicating their importance in Atlantic salmon spermatogenesis.

## 3.3. Plausible Sources and Functions of Seminal Plasma miRNAs

So far, no study has clarified the source and role of miRNAs in the seminal plasma. miRNAs can be present in the seminal plasma because they are leftovers after spermatogenesis, which is an intricate process involving cell proliferation and differentiation, as well as expulsion of cytoplasm content, and cell membrane disintegration [51]. Many miRNAs that have been implicated in mammalian spermatogenesis [17] were found in Atlantic salmon seminal plasma in the present study. Thus, seminal plasma miRNAs could be leftovers of the preceding spermatogenesis process. Accumulation during the enrichment of semen with organic and inorganic components from the blood plasma could hypothetically be another source of miRNAs in the seminal plasma. A study on the proteome in common carp (*Cyprinus carpio*) indicated that the majority of seminal plasma proteins were similar to blood plasma proteins [13]. However, in the present study, the profile of miRNAs in blood plasma versus seminal plasma was clearly dissimilar (Figure 6), with only one (miR-204) of 13 miRNAs showing concordant transcript levels, one miRNA (miR-451a) enriched in blood plasma, and six seminal plasma miRNAs absent in the blood plasma. This indicates that if the blood plasma is the source of seminal plasma miRNAs, the accumulation is rather selective than passive. However, no mechanism for such selective accumulation is known.

Most teleost fish have immobile spermatozoa in the seminal plasma, external fertilization, and a short fertilization window; all of this suggests that the seminal plasma RNAs are unlikely establish to an effective contact with an egg. Hence, the role of seminal plasma miRNAs may be restricted to maintaining a conducive environment for spermatozoa before the spermiation. Seminal plasma contains inhibitory factors that prevent the internalization of foreign nucleic acids [52,53,54,55]. It has been demonstrated that DNA-binding proteins at the sperm cell surface allow internalization of exogenous DNA in the absence of inhibitory factors in the seminal plasma [56]. Spermatozoa can also be transfected by RNA elements [57]. Exogenous DNA transfer to sperm has been shown in salmonids [58,59]. Seminal plasma miRNAs can protect spermatozoa against exogenous nucleic acid invasions, because they can exist in extra vesicular-free forms, associating with high-density lipoproteins or Argonaute proteins [60]. miRNAs have been demonstrated to act on targets outside the cell; for instance, intestinal epithelial cells release miRNAs to the gut to control the type and the level of microbiota [61]. Thus, it is possible that miRNAs bound to Argonaute proteins are among the inhibitory factors that facilitate degradation of exogenous RNAs in the seminal plasma, which is worthy of further investigations.

## 4. Materials and Methods

### 4.1. Ethics

Animal care and handling were done according to the guideline stated in Norwegian law of Animals in Research (The Norwegian Animal Protection Act no. 73 of 20 December 1974, Section 20–22, amended 19 June 2009).

### 4.2. Animals, Sperm Collection, and Fractionation

For small RNA sequencing experiments, sperm samples were collected from three sexually mature males from AquaGen Atlantic salmon breeding facility (AquaGen AS, Trondheim, Norway) by applying a gentle abdominal massage. The first portion of the milt was avoided to reduce the possibility of contamination with urine. Ziploc bags with sperm samples were placed on a crushed ice (0–2 °C) in an insulated polystyrene box and shipped overnight to Nord University, Bodø, Norway, for further processing. Upon arrival, sperm samples were checked for their quality under the microscope [62]. Special emphasis was given to determine whether there was contamination with exogenous cells, such as blood or other somatic cells, or bacteria. Each sperm sample was thoroughly screened through visual observation under the microscope (minimum five sub-samples, each screened in multiple view areas) for the presence of exogenous material. No visual contamination was found. Part of each sperm sample was fractionated to spermatozoa and seminal plasma. The separation was performed by centrifugation of 15 mL of whole semen at 6000 rpm for 30 min at 4 °C. After centrifugation, 6 mL of the upper clear layer of the supernatant was transferred to a new tube. To eliminate the contamination of seminal plasma with spermatozoa, the lower part of the supernatant was avoided. Absence of spermatozoa in the seminal plasma was checked under the microscope. The interphase between seminal plasma and the spermatozoa pellet was discarded. For the spermatozoa fraction, to remove seminal plasma leftovers, the spermatozoa pellet was diluted in an isotonic Hank’s balanced salt solution, centrifuged as above, and the supernatant was discarded. The procedure was repeated. The whole sperm samples, as well as corresponding fractions of spermatozoa and seminal plasma, were snap-frozen in liquid nitrogen and stored at −80 °C until RNA extraction.

For the quantitative reverse transcription PCR (RT-qPCR) experiment, sperm and blood samples were obtained from five adult males (>10 kg weight) of AquaGen origin. Seminal plasma and spermatozoa fractions were separated as described above. Two mL of blood were taken from the caudal vein using a needle and syringe, and centrifuged at 2000 rpm for 10 min to obtain blood plasma, which was snap frozen in liquid nitrogen and stored at −80 °C until RNA extraction.

### 4.3. RNA Extraction and Small RNA Sequencing

Total RNA was extracted from the whole semen, as well as its fractions: Spermatozoa and freeze-dried seminal plasma, using Trizol (Invitrogen, Carlsbad, California, USA) followed by extraction with chloroform, and ethanol precipitation at –80 °C overnight after adding glycogen (Life Technologies, Foster City, CA, USA) and 3M sodium acetate (Life Technologies). Precipitates were recovered by centrifugation at 12,000 rpm/4 °C for 1 h, washed with 750 µL 70% ethanol, and re-suspended in 20 µL RNase-free water. Quality of RNA was checked using bioanalyzer (Agilent Technologies, Waldbronn, Germany). Lack of contamination with exogenous cells in sperm samples was confirmed by the absence of 18S RNA and 28S RNA fractions [63]. RNA samples were stored at −80 °C until sequencing library preparation.

Sequencing libraries were prepared using an NEXTflex™ Illumina Small RNA Sequencing Kit v2 (Bioo Scientific, Austin, TX, USA) and sequenced on a NextSeq 500 (Illumina, San Diego, CA, USA) at the Genomics & Cell Characterization Core Facility, University of Oregon.

### 4.4. Data Analysis

Adapter sequences were trimmed using cutadapt [64] and sequence quality was checked using FastQC [65]. Sequences were mapped to Atlantic salmon miRNAs (www.miRBase.org) and to the salmon reference genome (ICSASG_v2 downloaded from www.SalmoBase.org) using Bowtie [66] with single mismatch and best alignment reporting. tRNAs were predicted using tRNAscan-SE [67] and conserved piRNAs were identified by mapping to zebrafish piRNA from piRBase [68].

Differential expression of miRNAs was determined for the whole semen, spermatozoa, and seminal plasma of Atlantic salmon by applying a negative binomial generalized linear model using DESeq2 [69] with the minimum threshold of 10 reads. We tested the differential accumulation for miRNAs that showed > 2-log fold change between the groups.

miRNA targets were predicted using Targetspy [70] on 3´UTRs extracted from genebank [71] and GO-term analysis for those targets genes was performed using Blast2Go [71].

### 4.5. In Situ Hybridization 

To determine the cellular origin of miRNAs in sperm, we performed in situ hybridization (ISH) on samples of mature (*n* = 3) and juvenile (2-year-old, weight 500 g, *n* = 3) testes fixed in Bouin’s solution (Sigma-Aldrich, Merck KGaA, Darmstadt, Germany) and embedded in paraffin. We also performed ISH on juvenile (*n* = 3) ovary sections for cross-sex comparisons. We used a double-DIG-labeled probe (5′-DIG and 3′-DIG) for miR-15-5p, miR-30d-5p, miR-92a-3p, miR-93a-5p, miR-202-5p, and miR-730-5p (miRCURY LNA™ Detection probe, Exiqon, Copenhagen, Denmark) according to the manufacturer’s instructions. In brief, embedded gonad sections (2.5 µm for juvenile testis, 5 µm for mature testis, and 5 µm for juvenile ovary) were incubated at 60 °C for 45 min to melt the paraffin, and stored overnight at 4 °C. On the next day, the sections were deparaffinized in xylene for 15 min and then re-hydrated through serially decreasing ethanol solutions. Targets were demasked using proteinase K (12 µg/mL) for 10 min. Sections were dehydrated in gradual ethanol solutions and dried. Denatured probes were diluted in 1x hybridization solution to a final concentration as shown in Appendix A. Sections were framed using Gene frame (Thermo Scientific, England, UK), and a total of 125 µL was applied on each slide. The slides were then covered with a Gene frame plastic cover slip and hybridized for 60 min at ~30 °C below the melting temperature of each probe (Appendix A). Slides were washed using 5×, 1×, and 0.2× SSC buffer (Sigma-Aldrich) at the hybridization temperature. Digoxin was recognized by a sheep anti-DIG-AP directly conjugated with alkaline phosphatase (AP) (Roche, Mannheim, Germany). The specimens were incubated for 60 min at room temperature. AP converted the applied substrate into a water and alcohol-insoluble dark-blue precipitate that appeared on slides after 2 h of incubation at 30 °C in the dark. The reaction was stopped, and slides were counterstained with 0.1% Nuclear Fast Red™ (Vector laboratories, CA, USA). After washing in tap water for 10 min and dehydration with sequential ethanol solutions, the slides were mounted using PERTEX^®^ (HistoLab, Goteborg, Sweden). The slides were examined under an Olympus BX51 microscope (Olympus, Tokyo, Japan) on the subsequent day and images were obtained using the Cell^B^ 27, build 1224 imaging software (Olympus).

### 4.6. Quantitative Reverse Transcription PCR (RT-qPCR)

In total, we performed RT-qPCR for selected 14 miRNAs using miRCURY LNA Universal RT microRNA PCR assays (Exiqon, Vedbaek, Denmark). First, 10 µL reverse transcription reaction was performed using the Universal cDNA synthesis kit with 20 ng total RNA in accordance with the manufacturer protocol. Reverse transcription thermocycling parameters were 42 °C for 60 min, 95 °C for 5 min. cDNA was diluted 1 in 80 and PCR was performed using Sybr Green mastermix (Exiqon) as per the manufacturer’s instructions with a 10 µL reaction volume using a Roche480 thermal cycler (Roche). PCR thermocycling conditions were as follows: 95 °C for 10 min, followed by 45 cycles of 95 °C for 10 s, 60 °C for 1 min, melt curve analysis performed between 60–95 °C for 15 min at a ramp-rate of 1.6 °C/s. The second derivatives method was used to calculate the quantification cycle (Cq) value. miR-429a was used as an endogenous reference, because it was expressed stably throughout the sample types (Appendix A). We performed pairwise comparisons using Wilcoxon rank sum test with Bonferroni correction, and *p*-value < 0.05 was considered as significant.

### 4.7. Assessment of Extracellular miRNAs Function

We constructed a minigene containing 6 targets sites for miR-202-5p and in vitro transcribed the sense and anti-sense of the construct using mMESSAGE mMACHINE^®^ kits (Invitrogen). We collected fresh sperm from Atlantic salmon and incubated in vitro transcribed RNAs in the seminal plasma or the whole semen, and monitored the RNA concentration and quality every 3 min for 30 min.

## 5. Conclusions

We characterized miRNAs in Atlantic salmon sperm fractions: Spermatozoa and seminal plasma. Differential expression of some miRNAs suggested that seminal plasma miRNA originated not only from spermatozoa. Testicular localization of selected miRNAs indicated distinctive expression in spermatozoa and germ cell supporting cells. As the miRNA profile in blood plasma showed low coherence with that from the seminal plasma, testicular supporting somatic cells are likely the source of miRNA enrichment in the seminal plasma. Their functions are unknown; we hypothesize their role in inhibition of exogenous nucleic acids in the sperm.

## Figures and Tables

**Figure 1 ijms-21-02723-f001:**
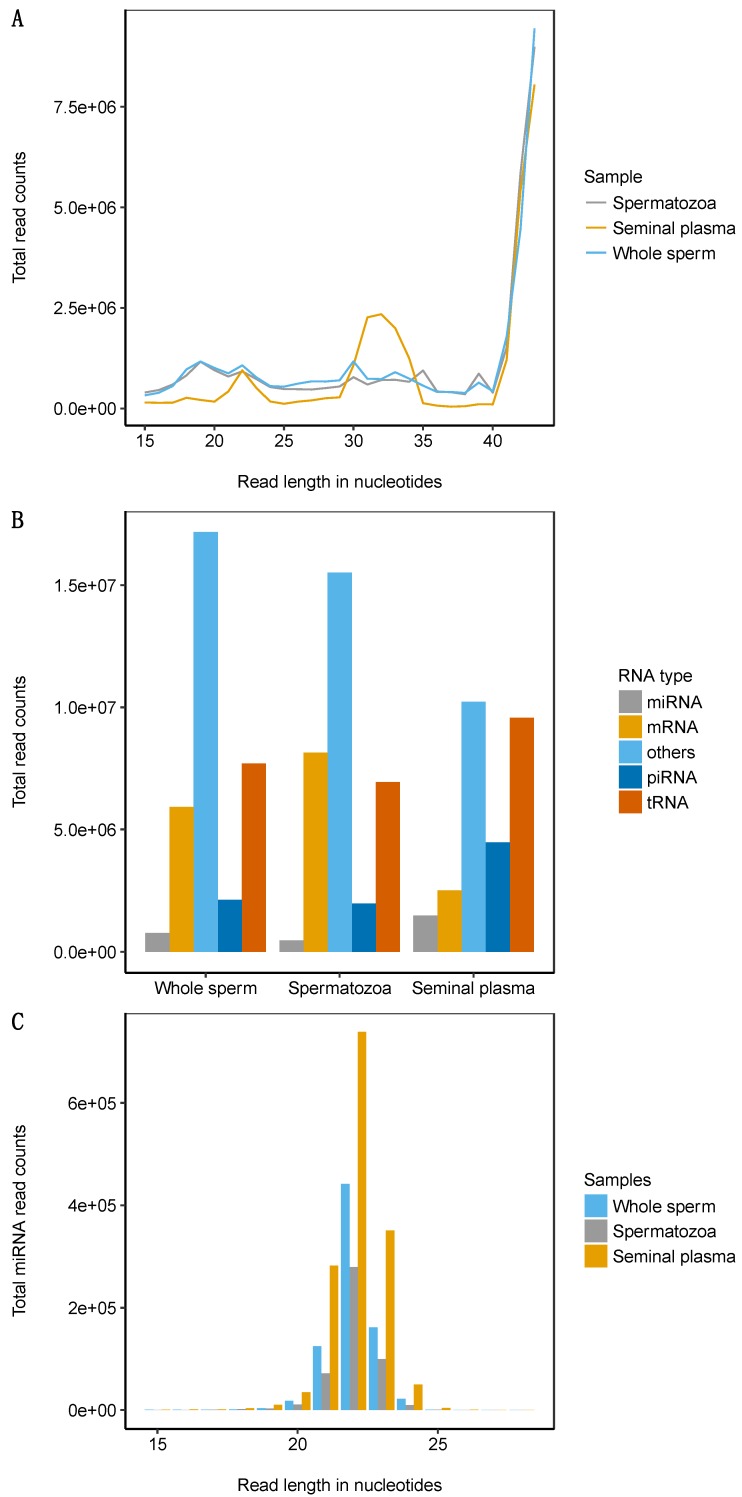
Major composition of small RNAs in Atlantic salmon sperm. (**A**) Sequence length distribution for each small RNA library of whole sperm, spermatozoa and seminal plasma. (**B**) Annotated RNA types. (**C**) micro RNA (miRNA) size distribution in the whole sperm, spermatozoa, and seminal plasma.

**Figure 2 ijms-21-02723-f002:**
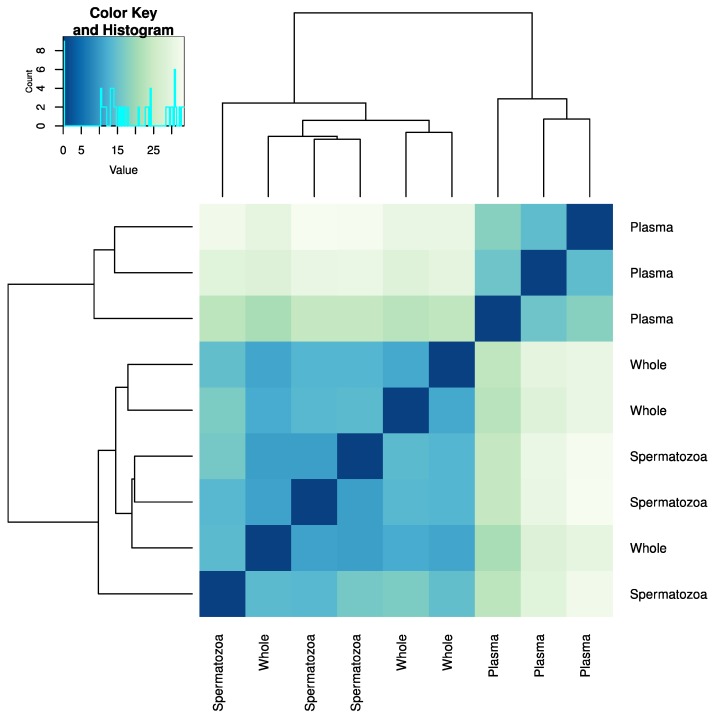
Cluster analysis of consistency of miRNA contents and abundances among the samples of the whole sperm and its fractionated spermatozoa and seminal plasma, obtained from three Atlantic salmon individuals. The dendrogram represents the similarity between samples.

**Figure 3 ijms-21-02723-f003:**
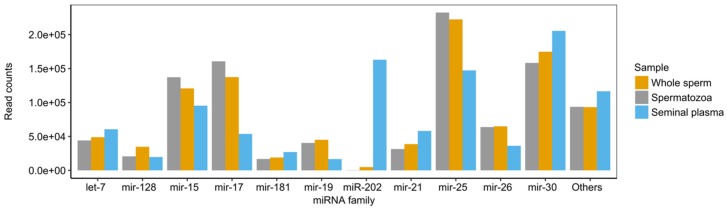
The most dominant miRNAs in the whole sperm and its fractions (spermatozoa and seminal plasma) of the Atlantic salmon.

**Figure 4 ijms-21-02723-f004:**
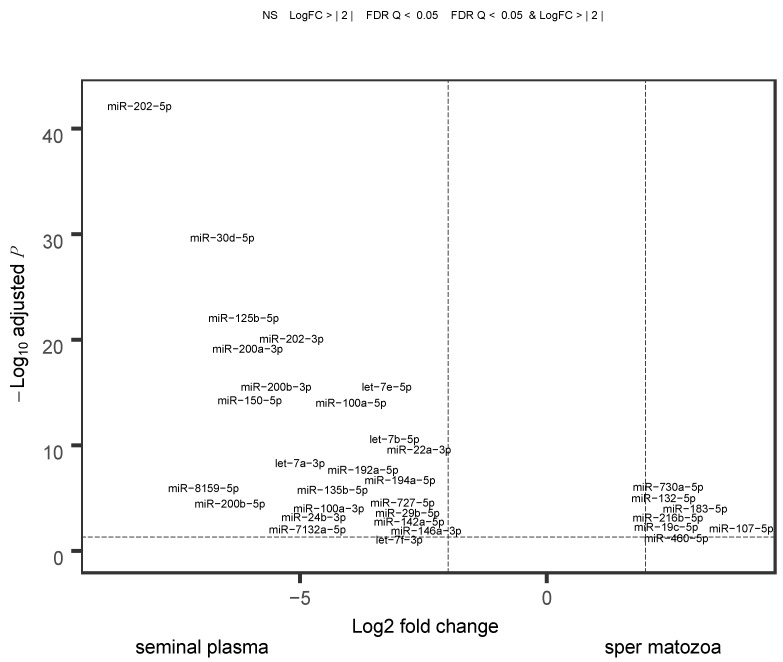
Differentially accumulated miRNAs between spermatozoa and seminal plasma fractions of Atlantic salmon sperm. Vertical and horizontal dashed lines show the fold change cut-off of 2 and adjusted *p*-value of 0.05, respectively. Red and labeled dots represent differentially accumulated miRNAs between the spermatozoa and seminal plasma fractions. FDR = false discovery rate; LogFC = logarithmic fold-change rate.

**Figure 5 ijms-21-02723-f005:**
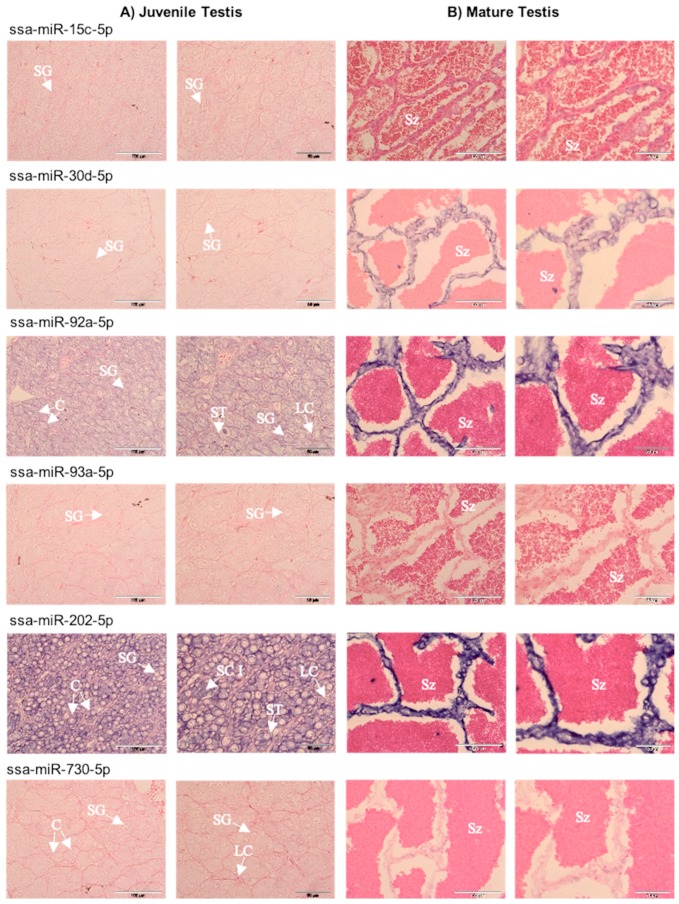
Localization of six miRNAs in sections of (**A**) juvenile and (**B**) mature Atlantic salmon testes, using in situ hybridization with locked nucleic acids (LNA )probes. Scale bars in the left column of each panel represent 100 µm, while the scale bars in the right column represent 50 µm. Abbreviations: C-cyst; LC-Leydig cell; ST- Sertoli cell; SG-spermatogonia; SC I- primary spermatocytes; Sz-spermatozoa.

**Figure 6 ijms-21-02723-f006:**
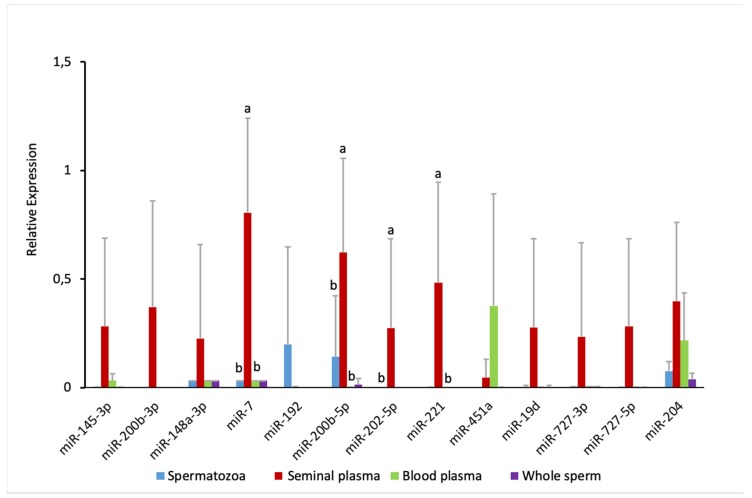
RT-qPCR for selected miRNAs in the blood plasma versus the whole sperm and its fractions (spermatozoa and seminal plasma). Relative expression values were obtained after normalizing to miR-429 expression. Average values are given (*n* = 6). Bars represent standard deviations. Different letters indicate significant differences at *p* < 0.05.

**Table 1 ijms-21-02723-t001:** Summary of sequencing depth (total reads, in millions), number of miRNA reads (in millions), and average read per size for three small RNA size groups, corresponding to putative miRNAs, putative Piwi-interacting RNAs (piRNAs), and other small RNAs, respectively. The average read/size is calculated using the number of total reads of a given size divided by a number of unique sequences in a given size.

Sample Type	Total Reads (min–max)	Total miRNA Reads (min–max)	Average Read/Size		
			15–26 nt	27–35 nt	36–40 nt
Whole sperm	31.2–37	0.6–0.9	48,681	70,808	50,636
Spermatozoa	29.7–37.2	0.4–0.6	52,191	62,011	51,635
Seminal plasma	20–35.5	0.4–2.7	19,026	17,126	8529

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
