# Peer review of "Heterogenic Origin of Micro RNAs in Atlantic Salmon (Salmo salar) Seminal Plasma"

_ijms, 2020, doi:10.3390/ijms21082723_

Round 1
Reviewer 1 Report
Authors present a quantitative analysis of the different miRNAs present in blood plasma, seminal plasma, sperm and whole semen in salmons. The study is interesting, since they carried out a complete analysis of all miRNA, providing also particular location for some of them, which were found also in testis.
What is not completely described is the impact of all these notions into the present literature and for this authors should better describe functional application of the whole study, emphasizing the limits and strengths of their results, and the weight of their study in the field.
Author Response
Reviewer #1:
Comments and Suggestions for Authors
Authors present a quantitative analysis of the different miRNAs present in blood plasma, seminal plasma, sperm and whole semen in salmons. The study is interesting, since they carried out a complete analysis of all miRNA, providing also particular location for some of them, which were found also in testis.
RESPONSE: Thank you very much
What is not completely described is the impact of all these notions into the present literature and for this authors should better describe functional application of the whole study, emphasizing the limits and strengths of their results, and the weight of their study in the field.
RESPONSE: We have added a paragraph (Page 10, Lines 7-14) to supplement the considerations of implications. We failed with a functional study to determine whether the miRNAs in seminal plasma have active, protective role, or they are just a leftover of spermatogenesis process (Page 10, Lines 15-18). We did not develop a discussion on the use of seminal plasma miRNAs as potential biomarkers of fertility and health conditions, as they are currently proposed in human medical diagnostics (reviewed by Robles et al. 2019. Non-coding RNA Research, 4, 54-62) because we do not believe such a use is relevant for fish. Rather, the next step is to elucidate the functional role of miRNAs in seminal plasma, which we postulated in the Discussion.
Reviewer 2 Report
The manuscript entitled “Heterogenic origin of microRNAs in Atlàntic (Salmo salar) seminal plasma is an excellent work. The manuscript is accurately written. The item is very interesting and innovative. Results are clear, showing interesting indications about the source of miRNAs and an interesting hypothesis of yo their role. This kind of studies are very important to understand the specie specific reproductive strategies and the possible relationships with other species phylogenetically far. Material and methods are detailed and under stable.
Author Response
Reviewer #2:
Comments and Suggestions for Authors
The manuscript entitled “Heterogenic origin of microRNAs in Atlàntic (Salmo salar) seminal plasma is an excellent work. The manuscript is accurately written. The item is very interesting and innovative. Results are clear, showing interesting indications about the source of miRNAs and an interesting hypothesis of yo their role. This kind of studies are very important to understand the specie specific reproductive strategies and the possible relationships with other species phylogenetically far. Material and methods are detailed and under stable.
RESPONSE: Thank you very much
Reviewer 3 Report
In general, the information given in the manuscript is ordered and clear. Just a few suggestion are given and some formal mistakes have been detected. They are listed down:
Suggestions:
- Try to increase the size of figure 4, specially miRNAs names. And correct “spermatozoa” legend.
- Give a brief explanation of what are “positive” or “negative controls” in section 2.3.
- Figure 5. Oictures and legends are too small. Full page figure?
- A description of the experiment described at P10L3-8 should be included in the m+m section.
- Explain better why ISH positives are found both in germinal and somatic cells in immature gonad, but it changes when the gonad maturation arrives.
- Section 3.3. Could miRNAs have an specific role during the final sperm maturation process (capacitation)?
Minor mistakes:
P1L37 [8,9]
P2L2 microvesicles?
P2,L31-37 (and rest of the manuscript). Check journal format for the use of “%”. Usually, it should be written linked to the number (i.e.: 10% and not 10 %), and using a single symbol if different values are cited (i.e.: 5 and 10%).
P6,L15 to spermatogonia?, at? in?
P6L21 In situ hibridization (ISH). First time in the text.
P10L19 Check font´s size.
P11L4 intrincated?
P11L16 …for such selective…?
P12 L1 vs L12 and 20. Check journal format to write thousands separation (i.e. 6000 or 6,000?)
P12L18 -80 (check symbol used for Celsius degrees in this and following page).
P12L30 Check font´s size.
P12L36 Delete space before “of 10…”.
References section. Lots of mistakes (short/long journal names?; species names must be written in italics; avoid using capitals in every word of the paper´s titles; etc.). Please follow the journal rules.
Author Response
Reviewer #3:
Comments and Suggestions for Authors
In general, the information given in the manuscript is ordered and clear. Just a few suggestion are given and some formal mistakes have been detected. They are listed down:
Suggestions:
Try to increase the size of figure 4, specially miRNAs names. And correct “spermatozoa” legend.
RESPONSE: The Figure 4 has been enlarged to the page limits, and the font size in annotated miRNAs has been enlarged. We could not find any error in “spermatozoa” legend.
Give a brief explanation of what are “positive” or “negative controls” in section 2.3.
RESPONSE: The information has been provided (Page 6, Lines 10-15).
Figure 5. Oictures and legends are too small. Full page figure?
RESPONSE: The Figure 5 display has been changed to allow increase in size of the pictures.
A description of the experiment described at P10L3-8 should be included in the m+m section.
RESPONSE: It is now included (new subsection 4.7, Page 13).
Explain better why ISH positives are found both in germinal and somatic cells in immature gonad, but it changes when the gonad maturation arrives.
RESPONSE: Generally, it can be a technical issue (signal presentation in certain type of material) or biological (change of expression pattern, quite common in the course of development). Here, regardless of the latter possibility (biology), we think this feature is at least partially related to the technical issues. Please look at the Supplementary Figure 5: the positive control shows no/weak signal in spermatozoa, while it should be rather strong. The reason is most likely in the quality of the material (small, loose cells) disabling proper signal exhibition. We have performed this experiment several times, changing reaction conditions, but the results were the same.
We have addressed it on Page 6 (Lines 10-15 and Lines 23 – 3 on the next page)
Section 3.3. Could miRNAs have an specific role during the final sperm maturation process (capacitation)?
RESPONSE: Yes, they could. Normally, there is a reason for accumulation of regulatory molecules in an organ/tissue. However, it would be quite tricky to examine: One would need to build up a system allowing non-biased sampling of purified stages throughout sperm cell development, in the same time collecting supporting somatic cells and/or fluids. Other possibility, more realistic, could be by means of reverse genetics, to elucidate how the quantitative alteration of a chosen component (e.g. miRNA) affect the final sperm manturation.
Minor mistakes:
P1L37 [8,9]
P2L2 microvesicles?
P2,L31-37 (and rest of the manuscript). Check journal format for the use of “%”. Usually, it should be written linked to the number (i.e.: 10% and not 10 %), and using a single symbol if different values are cited (i.e.: 5 and 10%).
P6,L15 to spermatogonia?, at? in?
P6L21 In situ hibridization (ISH). First time in the text.
P10L19 Check font´s size.
P11L4 intrincated?
P11L16 …for such selective…?
P12 L1 vs L12 and 20. Check journal format to write thousands separation (i.e. 6000 or 6,000?)
P12L18 -80 (check symbol used for Celsius degrees in this and following page).
P12L30 Check font´s size.
P12L36 Delete space before “of 10…”.
RESPONSE: Thank you very much for spotting them out. We have corrected these errors.
References section. Lots of mistakes (short/long journal names?; species names must be written in italics; avoid using capitals in every word of the paper´s titles; etc.). Please follow the journal rules.
RESPONSE: We have corrected the References.